# ONEST (Observers Needed to Evaluate Subjective Tests) Analysis of Stromal Tumour-Infiltrating Lymphocytes (sTILs) in Breast Cancer and Its Limitations

**DOI:** 10.3390/cancers15041199

**Published:** 2023-02-14

**Authors:** Bálint Cserni, Darren Kilmartin, Mark O’Loughlin, Xavier Andreu, Zsuzsanna Bagó-Horváth, Simonetta Bianchi, Ewa Chmielik, Paulo Figueiredo, Giuseppe Floris, Maria Pia Foschini, Anikó Kovács, Päivi Heikkilä, Janina Kulka, Anne-Vibeke Laenkholm, Inta Liepniece-Karele, Caterina Marchiò, Elena Provenzano, Peter Regitnig, Angelika Reiner, Aleš Ryška, Anna Sapino, Elisabeth Specht Stovgaard, Cecily Quinn, Vasiliki Zolota, Mark Webber, Sharon A. Glynn, Rita Bori, Erika Csörgő, Orsolya Oláh-Németh, Tamás Pancsa, Anita Sejben, István Sejben, András Vörös, Tamás Zombori, Tibor Nyári, Grace Callagy, Gábor Cserni

**Affiliations:** 1TNG Technology Consulting GmbH, Király u. 26., 1061 Budapest, Hungary; 2Discipline of Pathology, Lambe Institute for Translational Research, School of Medicine, University of Galway, H91 TK33 Galway, Ireland; 3Pathology Department, Atryshealth Co., Ltd., 08039 Barcelona, Spain; 4Department of Pathology, Medical University of Vienna, Währinger Gürtel 18-20, 1090 Vienna, Austria; 5Division of Pathological Anatomy, Department of Health Sciences, University of Florence, 50134 Florence, Italy; 6Tumor Pathology Department, Maria Sklodowska-Curie National Research Institute of Oncology, Gliwice Branch, 44-102 Gliwice, Poland; 7Laboratório de Anatomia Patológica, IPO Coimbra, 3000-075 Coimbra, Portugal; 8Laboratory of Translational Cell & Tissue Research and KU Leuven, Department of Imaging and Pathology, Department of Pathology, University Hospitals Leuven, University of Leuven, Oude Market 13, 3000 Leuven, Belgium; 9Unit of Anatomic Pathology, Department of Biomedical and Neuromotor Sciences, University of Bologna, Bellaria Hospital, 40139 Bologna, Italy; 10Department of Clinical Pathology, Sahlgrenska University Hospital, 41345 Gothenburg, Sweden; 11Department of Pathology, Helsinki University Central Hospital, 00029 Helsinki, Finland; 12Department of Pathology, Forensic and Insurance Medicine, Semmelweis University Budapest, Üllői út 93, 1091 Budapest, Hungary; 13Department of Surgical Pathology, Zealand University Hospital, 4000 Roskilde, Denmark; 14Department of Pathology, Riga Stradins University, Riga East Clinical University Hospital, LV-1038 Riga, Latvia; 15Unit of Pathology, Candiolo Cancer Institute FPO-IRCCS, 10060 Candiolo, Italy; 16Department of Medical Sciences, University of Turin, 10126 Turin, Italy; 17Department of Histopathology, Cambridge University Hospitals National Health Service (NHS) Foundation Trust, Cambridge CB2 0QQ, UK; 18National Institute for Health Research Cambridge Biomedical Research Centre, Cambridge CB2 0QQ, UK; 19Diagnostic and Research Institute of Pathology, Medical University of Graz, 8010 Graz, Austria; 20Department of Pathology, Klinikum Donaustadt, 1090 Vienna, Austria; 21The Fingerland Department of Pathology, Charles University Medical Faculty and University Hospital, 50003 Hradec Kralove, Czech Republic; 22Pathology Department, Herlev University Hospital, DK-2730 Herlev, Denmark; 23Department of Histopathology, Irish National Breast Screening Programme, BreastCheck, St. Vincent’s University Hospital and School of Medicine, University College Dublin, D04 T6F4 Dublin, Ireland; 24School of Medicine, University College Dublin, D04 V1W8 Dublin, Ireland; 25Department of Pathology, School of Medicine, University of Patras, 26504 Rion, Greece; 26Department of Pathology, Bács-Kiskun County Teaching Hospital, 6000 Kecskemét, Hungary; 27Department of Pathology, University of Szeged, 6720 Szeged, Hungary; 28Department of Medical Physics and Informatics, University of Szeged, 6720 Szeged, Hungary

**Keywords:** ONEST, observers needed to evaluate subjective tests, TILs, sTILs, tumour-infiltrating lymphocytes, triple-negative, breast cancer, reproducibility, international immuno-oncology biomarker working group, European Working Group for Breast Screening Pathology

## Abstract

**Simple Summary:**

Tumour-infiltrating lymphocytes (TILs) reflect the host’s response against tumours. TILs have a strong prognostic effect in the so-called triple-negative (oestrogen receptor, progesterone receptor, and human epidermal growth factor receptor-2 negative) subset of breast cancers and predict a better response when primary systemic (neoadjuvant) treatment is administered. Although they are easy to assess, their quantitative assessment is subject to some inter-observer variation. ONEST (Observers Needed to Evaluate Subjective Tests) is a new way of analysing inter-observer variability and helps in estimating the number of observers required for a more reliable estimation of this phenomenon. This aspect of reproducibility for TILs has not been explored previously. Our analysis suggests that between six and nine pathologists can give a good approximation of inter-observer agreement in TIL assessments.

**Abstract:**

Tumour-infiltrating lymphocytes (TILs) reflect antitumour immunity. Their evaluation of histopathology specimens is influenced by several factors and is subject to issues of reproducibility. ONEST (Observers Needed to Evaluate Subjective Tests) helps in determining the number of observers that would be sufficient for the reliable estimation of inter-observer agreement of TIL categorisation. This has not been explored previously in relation to TILs. ONEST analyses, using an open-source software developed by the first author, were performed on TIL quantification in breast cancers taken from two previous studies. These were one reproducibility study involving 49 breast cancers, 23 in the first circulation and 14 pathologists in the second circulation, and one study involving 100 cases and 9 pathologists. In addition to the estimates of the number of observers required, other factors influencing the results of ONEST were examined. The analyses reveal that between six and nine observers (range 2–11) are most commonly needed to give a robust estimate of reproducibility. In addition, the number and experience of observers, the distribution of values around or away from the extremes, and outliers in the classification also influence the results. Due to the simplicity and the potentially relevant information it may give, we propose ONEST to be a part of new reproducibility analyses.

## 1. Introduction

Tumour-infiltrating lymphocytes (TILs) are a reflection of antitumour immunity. Different compartments and populations are recognised; for breast carcinomas, stromal lymphocytes have been accepted as the most practically assessable compartment of TILs, and their quantity correlates with that of intra-epithelial TILs [1]. On the basis of meta-analyses, stromal TILs (sTILs) have been proven to be predictive of the response to neoadjuvant chemotherapy [2] and to be associated with better prognosis after adjuvant treatment of triple-negative breast carcinomas (TNBCs) [3]. TILs have also been linked to the rare phenomenon of spontaneous regression in TNBC [4]. The accumulated data on the value of TILs have matured enough to recommend this biomarker for implementation in daily routine [5].

However, there are a number of other events (e.g., necrosis or previous biopsy) that lead to the accumulation of inflammatory cells, and these have been taken into consideration when defining the rules for quantifying the amount of sTILs relevant for antitumour immunity. This has led to the formulation of guidelines recommending that sTILs should be evaluated as the average proportion of the stromal area occupied by TILs, including both lymphocytes and plasma cells. In the assessment, the total stromal area excludes areas of regressive hyalinisation, necrosis, and previous needle biopsy sites. Mononuclear cells around in situ carcinoma and normal structures should also be excluded, and all estimations should be restricted to the tumour area [6]. A later addendum suggested that the invasive front (1 mm at the edge of the tumour) should also be included [7]. The human brain tries to simplify things; therefore, the rules for quantifying sTILs predispose this biomarker to being poorly reproducible. Nevertheless, good reproducibility was documented by the International Immuno-oncology Biomarker Working Group (IIOBMWG) after the introduction of a direct online feedback software helping in the calibration of sTIL percentages in pre-selected fields of view (FOVs) [8].

Members of the European Working Group for Breast Screening Pathology (EWGBSP) have also assessed the reproducibility of scoring sTILs on digitised needle core biopsy specimens using the same performance-improving online tool that was used for training by Denkert et al. [8,9] and found moderate reproducibility for biopsy specimens (intraclass correlation coefficient, ICC 0.634, 95% CI 0.539–0.735) but good reproducibility for selected triplets of FOVs (ICC 0.798, 95% CI 0.727–0.864) [10]. In the present work, we use the same data to perform an ONEST (Observers Needed to Evaluate Subjective Tests) analysis of sTILs.

ONEST is a recently developed method that complements inter-observer agreement studies by helping to estimate the number of observers required for a reliable estimation of reproducibility [11]. ONEST uses 100 randomly selected permutations of all participating pathologists (observers or raters) and plots the overall percent agreement (OPA) values for an increasing number of observers, looking for the worst (lowest) curve to reach a plateau, beyond which an increasing number of observers does not have a substantial effect on agreement [11,12,13]. Additionally, ONEST has been recognised to be valuable as a visual complement to demonstrate the degree of reproducibility of subjectively evaluated parameters such as oestrogen receptor (ER) quantification, Ki-67 labelling, or histological grade, as well as the difference between observers and how these compare to the overall percent agreement (OPA) of all observers [12,13]. The aim of this study is to evaluate sTIL quantification using ONEST and to estimate the number of observers needed for a reliable evaluation of its reproducibility. The relatively large number of observers in our previous study [9] allows for a better evaluation of ONEST itself as a method.

## 2. Materials and Methods

We used anonymised results from the EWGBSP analysis of reproducibility [9]. In that study, 23 pathologists assessed 49 core needle biopsies from TNBCs in circulation 1 (C1), and 14 pathologists, as a subset, assessed both C1 (this subset of C1 denoted as C1s) in addition to 3 pre-selected digital FOVs of the same 49 cases with different labels to prevent comparisons (C2). The corresponding author of this previous study (Grace Callagy) has released the sTIL percentage values reported by the 23 and 14 participants for each case in a tabulated format, with rows representing cases and columns representing one or the other observer, and these values were used for the ONEST analyses of C1 and C2, respectively. There were 2 missing values in all circulations (C1, C1s, and C2) which were replaced by mean sTIL percentages rounded to the closest integer. For the ONEST analysis, as per the introduction of the method and its subsequent uses [11,12,13], 100 randomly selected permutations were selected for the values of the ONEST plots. Four selected cut-offs were used to define categories: <60% vs. ≥60%, e.g., [14], and <50% vs. ≥50%, e.g., [15,16], to match two different definitions of lymphocyte-predominant breast cancers, which are the likeliest responders to neoadjuvant treatment [6]; <30% vs. ≥30% to match a cut-off proposed for a strong prognostic role in the adjuvant setting [3]; and 0–20%, 21–49%, and ≥50% to match a three-tiered classification used in the IIOBWG ring studies [8].

In a previous study, 9 pathologists assessed the ER, the progesterone receptor (PR) status, Ki67 labelling [12], and histological grade [13] of breast cancers in 50 core needle biopsies and 50 resection specimens represented on a full-face glass slide for each case. While assessing these parameters, the participants were also asked to document sTILs based on the IIOBMWG recommendations [6,7], which are also part of the Hungarian recommendation [17,18]. These results have never been analysed previously and were also used for a separate ONEST analysis as circulation 3 (C3).

A full ONEST plot includes all OPA values per increasing number of observers for the 100 randomly obtained permutations of observers, i.e., it represents 100 OPA curves (OPACs), each representing the OPA values of a given permutation (Figure 1A). We also introduced a simplified ONEST plot, which includes only the maximum OPA values (maximum curve—best scenario), the minimum OPA values (minimum curve—worst scenario), and a median value curve. The maximum and minimum curves do not necessarily represent an OPAC from the 100 randomly selected permutations, but they obviously coincide with an OPAC from all possible permutations. Figure 1A and 1B compare the full and simplified ONEST plots of the same entity studied. The ONEST value is the integer from axis *x* (the number of pathologists), which reflects the minimum curve OPA value beyond which there is no more relevant decrease in OPA values with further increase in observers. Bandwidth is defined as the difference between the highest and lowest OPA values with 2 pathologists assessing sTILs, i.e., this is the difference in OPA of the maximum and minimum curves with 2 observers. Finally, OPA(n) is the OPA value for all observers, the percentage of cases upon which all assessing observers agree. Good reproducibility implies a high OPA(n), a low ONEST value, and narrow bandwidth, whereas the opposite is true for poor reproducibility. The worst scenario is when OPA(n) = 0, i.e., there are no cases on which all observers agree. This latter scenario is unacceptable for biomarker studies or subjective tests on relevant issues in general and should be remedied by improving reproducibility or dropping the test and substituting it with a better one. An open-source software designed by the first author for randomly selecting 100 permutations from all possible ones and making a basic ONEST analysis is available at github.com (accessed on 12 November 2022) [19].

For the analysis with a cut-off value of <50% vs. ≥50% sTILs, ONEST analyses were repeated 3 times (3 random selections of 100 permutations in which the chances of identical permutations are practically nil), and the minimum curves obtained were compared by means of the Kruskal–Wallis test. In the original series, two pathologists (numbers 7 and 20) substantially diverged in their opinions from the rest of the group in C1, whereas two pathologists (numbers 4 and 13) diverged from others in C2. To test the influence of these divergently classifying pathologists, 3 and 3 ONEST plots for the same cut-off values (<50% vs. ≥50%) were also generated after the removal of results by these observers, and the ONEST values were determined from all plots. The ONEST values obtained with or without the deviant classifiers were compared by means of the two sample Wilcoxon rank sum test. Statistical analyses were performed in Excel with the Real Statistics Add-Ins [20] and STATA Software version 17.0 (StataCorp LP, College Station, TX, USA).

## 3. Results

The results of the C1 were selected to be represented by the ONEST plots in Figure 1. Readings of this and other ONEST analyses from C2 and C3 are represented in Table 1. With different approaches, pathologists, and numbers of pathologists, the ONEST values varied between 2 and 11 (Table 1). There were two pathologists, in both circulations C1 and C2, who substantially deviated from the overall average ratings; separate ONEST analyses were also performed without these participants. Not surprisingly, not only did the OPA(n) values increase, but the bandwidth became smaller, and the ONEST values decreased. With the exception of the C1 (n = 23 pathologists) for the <50% vs. ≥50% and the <30% vs. ≥30% categorisations, the ONEST values were not greater than 9; the number of pathologists involved in the C3 yielded the best OPA(n) values, i.e., the best reproducibility (Table 1).

One of the sTIL categorisations was used to test the ONEST plots. Three random selections of 100 permutations were compared with the Kruskal–Wallis test for the chosen (<50% vs. ≥50%) sTIL categorisation for C1, C1 without the two substantially divergent raters, C1s, C2, and C2 without the two substantially divergent raters. Although sometimes there was a small shift in the ONEST and other values, these permutations were not statistically different with regard to the minimum curves; their p-values were 0.937, 0.271, 0.877, 0.855, and 1, respectively (Figure 2).

Furthermore, the three random permutations from C1 and C1 without the outlying classifiers; C2 and C2 without the outlying classifiers; and finally, C1 and C2, were also compared with the Wilcoxon rank sum test for the ONEST values that could be derived from them, and this demonstrated significant differences (*p* = 0.046, *p* = 0.034, and *p* = 0.043, respectively) for each of these comparisons.

## 4. Discussion

ONEST is a recently described additional analysis that can complement reproducibility studies [11,12,13]. Although it was introduced to estimate the minimum number of observers required to provide a reliable estimate of the reproducibility of a given classification [11], it also gives a visual impression of how much agreement is reached when categorising items into predefined classes and the difference one can expect between two observers. However, as a complementary tool, ONEST is not independent of the studied “population” and the observers.

It is generally accepted that two-tiered classifications are more reproducible than those with more than two categories, e.g., [21]. This also applies to ONEST, as reported for PD-L1 [9] and Ki67 [10], and this is also supported by our analysis of the three-tiered classification in the present study, which demonstrated the worst OPA(n) values in nearly all circulations (Table 1).

Although our attempt to analyse the data without the two observers who substantially deviated from the majority opinion resulted in “improved” results in both C1 and C2 (i.e., greater OPA(n), narrower bandwidth, and lower ONEST values), the analyses without these outliers may not reflect real-life assessments. It is well accepted that populations are generally described with their average values of measurable things, but they also have members that are above and below the average. Therefore, if one wishes to estimate the real-life performance of a classification, all raters, and not only the best raters, should be included in the analysis.

Reproducibility is also dependent on the distribution of the parameter being evaluated in the cases. While assessing three nuclear immunostains for ER, PR, and Ki67 in a different study, we found that using the same cut-off values for all three biomarkers resulted in different reproducibility and ONEST estimations [12]. This was explained by the difference in the number of cases close to or away from the extreme values (0% and 100%). Most values for ER staining were in the 90–100% or 0% range, whereas PR values showed more divergence, Ki67 scores were distributed over a wider range, and ONEST values increased in a respective manner. This phenomenon is likely to be the most important contributor to the surprisingly good results observed for the C3 circulation in the present study (Table 1). Indeed, in C3, there were only 45/900 ratings for sTILs ≥ 50% involving 8/100 cases.

The homogeneity versus heterogeneity of the entity being observed also influences reproducibility, and this is substantiated by earlier studies. ER staining is generally more homogeneous than Ki67, as reflected in the lower inter-observer agreement for the latter [12]. On the other hand, sTILs often have a heterogeneous distribution, making it more difficult to assess the overall average distribution. This phenomenon, i.e., heterogenous distribution, was identified as the main contributor to the weaker reproducibility for some cases in our previous study [9] and was also reported by others [22]. Scoring preselected FOVs (C2) eliminates the variability associated with the observers selecting the areas to score in the case of heterogeneously distributed sTILs and results in substantially better reproducibility (ICC for absolute sTILs with preselected FOVs vs. the case when observers had selected their FOVs to be assessed: 0.798 vs. 0.634) [9]. This improved reproducibility was also reflected by key values of ONEST plot analyses: higher OPA(n) values, lower bandwidth, and lower ONEST values in C2 vs. C1 for all categorical classifications.

The number of observers may also influence reproducibility and ONEST plots. For example, C2 versus C1, without the discordant raters (with 12 observers of the former all included in the 21 of the latter), resulted in different OPA(n) values (82% vs. 61% agreement for, e.g., sTILs ≥ 50% or fewer). The number of observers also greatly impacts the number of possible permutations, being 2.585 × 10^22^ for C1 (n = 23), 87,178,291,200 for C2 (n = 14), and “only” 362,880 for C3 (n = 9). In a previous study, also with nine observers [12], we verified that the minimum curve of the 100 randomly selected permutations does not significantly differ from the minimum OPAC of all permutations. In the present analysis, three random ONEST plots were examined for all circulations with one of the cut-offs (<50% vs. the ≥50%), and no significant difference was found between their minimum curves. This is also reflected in Figure 2, in which the minimum (and the maximum and median) curves of the three plots substantially overlap with each other. Despite this, there were minor alterations in the bandwidths and ONEST values from the three analyses of the same datasets. This leads us to conclude that even ONEST readings are just estimations and might have a range, but depending on how close the ONEST value is to 2, we can estimate how a reproducibility study with a low number of participants may reflect real-life performance for the test in question. An early study of TILs with 99 cases suggested an 85% (95% CI, 76% to 91%) agreement with no more than a 10% difference in absolute sTIL ratings between two observers [23]. Kojima and colleagues reported an 81% agreement between two observers when classifying sTILs into three categories in 129 cases [24]. A report on 100 cases and >90% mean pairwise agreement on sTILs, by any of six pathologists, with a seventh pathologist serving as the main reviewer for a study, also suggests excellent reproducibility [25]. However, Figure 1A clearly shows that two observers randomly selected from a pool of observers or pairwise comparisons may have minimal discrepancies or no discrepancy at all, but the bandwidth may be much wider than this. Four pathologists also achieved a good agreement scoring sTILs in 121 cases [26] and substantial agreement in 75 cases [27], but Table 1 suggests that this number is still prone to underestimating real-life conditions. Certainly, two observers [23,24,28,29] do not accurately reflect inter-observer agreement [11], and most readings from the ONEST plots (Table 1) with a different number or quality of readers suggest that between 6 and 11 readers are required for a reasonable estimation of inter-observer agreement.

As a limitation, ONEST analyses can only be performed for categorical classifications. Agreement for scoring some markers (e.g., sTILs) as a continuous variable is generally better than the agreement observed using categories defined by given cut-off values [30]. On the other hand, therapeutic decisions are generally made using cut-off values for a biomarker.

Finally, after considering the factors influencing the reproducibility of a subjective test, such as scoring sTILs in breast cancer, it is the case that other variables (e.g., number and experience of observers, distribution of the cases around or away from the extremes, and heterogeneity between fields to assess) also influence ONEST analyses and the ONEST values. Therefore, we can state that two to four observers are certainly not sufficient to reflect the actual inter-observer agreement for evaluating sTILs in breast cancer, but between 6 and 11 observers would be sufficient. The studies by the IIOBMWG largely fulfil this requirement, and their reported values of good reproducibility should be considered reliable [8]. Notwithstanding, the finding that our group, also with a sufficient number of pathologists, was only able to match their high ICC values when scoring sTILs on preselected FOVs, but not when full digital slides were scored, clearly means that factors other than the number of observers contribute to reproducibility [11]. This is also substantiated by another study involving 41 cases of digitised core needle biopsies scored by 40 pathologists, where the ICC values ranged between −0.376 and 0.947, with a mean of 0.659 [31]. In addition to applying methods such as ONEST, the development of tools that can quantify other contributors to lower reproducibility will be useful in the design of reproducibility studies. Due to its simplicity and the data it gives, we also propose that an ONEST analysis could be a part of reproducibility studies to explore the reliability of the results presented or published previously, as not all reports satisfy the suggested minimum number of observers to reach the best possible conclusions. However, the limitations described in the present article must be kept in mind.

## 5. Conclusions

The reproducibility of sTIL assessments in breast cancer has been examined in several studies. Our results using ONEST indicate that between six and nine observers are expected to give a good estimate of inter-observer variability, and studies involving fewer than these numbers may overestimate agreement between observers. As sTIL evaluation becomes part of daily practice [5], efforts to characterise factors interfering with the reproducibility of scoring are welcome.

## Figures and Tables

**Figure 1 cancers-15-01199-f001:**
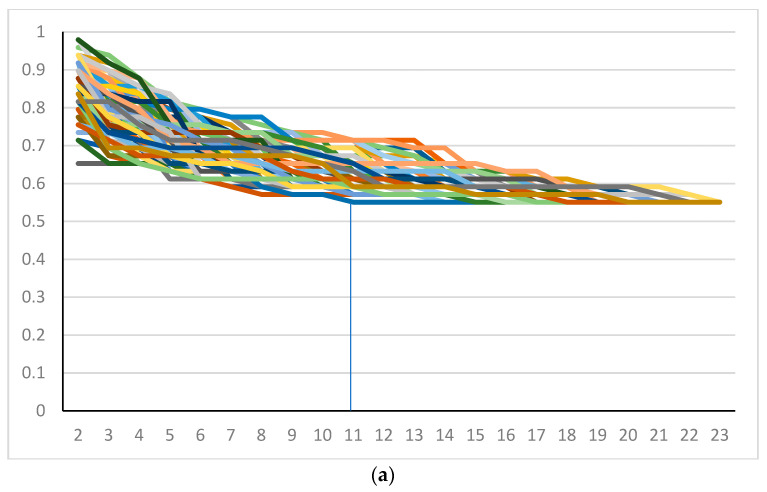
ONEST plots of different cut-off values for 23 pathologists. (**a**) Full and (**b**) simplified ONEST plots for the 49 cases assessed by 23 pathologists for a cut-off of <50% vs. ≥50% sTILs. (**c**–**e**) Simplified ONEST plots for further cut-off values studied: (**c**) <60% vs. ≥60%; (**d**) <30% vs. ≥30%; and (**e**) <20%, 21–50%, >50%. Readings from the plots are included in Table 1. OPA (n = 23) values are the OPA values at the right side of the plots and reflect the proportion of cases with full agreement. ONEST values correspond to the number of observers on the *x*-axis, where the minimum curve levels off, and no substantial decrease is noted with further increase in the number of observers (this is highlighted by vertical segments between the *x*-axis value and the minimum curve). The bandwidth of the ONEST plot is visualised on the left side of the plot as the difference between the maximum and the minimum curves with 2 observers; this is the largest difference in agreement between two observers.

**Figure 2 cancers-15-01199-f002:**
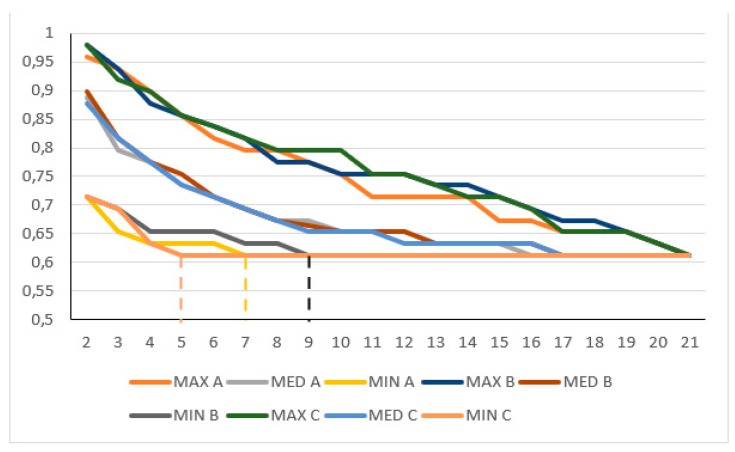
Partly overlapping simplified ONEST plots of 3 randomly selected 100 permutations (A, B, and C) for the <50% sTIL or more classification in C1 circulation without the two divergent classifiers; this example showed the lowest *p*-value in the Kruskal–Wallis test. Note: the *y*-axis only represents values between 0.5 and 1; despite not being statistically significantly different, the 3 randomly selected ONEST plots of 100 permutations yield 3 different ONEST values: 7 (A), 9 (B), and 5 (C) (to ease reading of the values, these are highlighted by vertical dashed segments between the *x*-axis value and the minimum curves), whereas the bandwidth is very similar (0.245 A, 0.265 B and C), and by definition, the OPA(21) value is identical (0.612). MAX: maximum curve; MED: median curve; MIN: minimum curve.

**Table 1 cancers-15-01199-t001:** ONEST analyses of different circulations and cut-off values of sTILs.

<50% vs. ≥50%	C1	C1 without Divergent Raters 7 and 20	C1s	C2	C2 without Divergent Raters 4 and 13	C3
n	23	21	14	14	12	9
OPA(n)	0.551	0.612	0.571	0.776	0.816	0.89
Bandwidth	0.327	0.245	0.265	0.184	0.143	0.07
ONEST	11	7	8	6	3	6
<60% vs. ≥60%						
n	23	21	14	14	12	9
OPA(n)	0.612	0.796	0.612	0.796	0.837	0.91
Bandwidth	0.327	0.286	0.612	0.163	0.163	0.07
ONEST	9	7	4	6	2	2
<30% vs. ≥30%						
n	23	21	14	14	12	9
OPA(n)	0.306	0.347	0.327	0.551	0.592	0.81
Bandwidth	0.408	0.306	0.306	0.306	0.204	0.09
ONEST	11	8	9	8	7	6
≤20%, 21–49%, ≥50%						
n	23	21	14	14	12	9
OPA(n)	0.163	0.204	0.408	0.408	0.449	0.74
Bandwidth	0.469	0.388	0.143	0.265	0.245	0.12
ONEST	8	7	7	5	6	6

C1: circulation 1 with 23 pathologists and 49 digital slides of core needle biopsy samples; C1s: subset of C1 with the 14 pathologists taking part in C2; C2: circulation 2 with 14 pathologists and 3 preselected fields of view of the 49 cases viewed in C1; C3: circulation 3 is independent from C1 and C2 and involves 9 pathologists assessing 100 cases, half from core needle biopsies and half from excision specimens. For further details, see the Materials and Methods section.

## Data Availability

Data are available from the corresponding author and will be released upon reasonable request.

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
