# Peer review of "ONEST (Observers Needed to Evaluate Subjective Tests) Analysis of Stromal Tumour-Infiltrating Lymphocytes (sTILs) in Breast Cancer and Its Limitations"

_cancers, 2023, doi:10.3390/cancers15041199_

Round 1
Reviewer 1 Report (Previous Reviewer 3)
This manuscript has been reviewed as cancers-2065476. Please add the point-to-point response as the cover letter. My previous comments were as below.
The authors want to evaluate stromal TILs (sTILs) quantification by ONEST and to estimate the number of observers needed for a reliable evaluation of its reproducibility. There are several problems:
1. What is the difference between this study and the previous works, especially the ONEST analyses, using open-source software developed by the first author? There must be enough differences.
2. The author used anonymized results from the EWGBSP analysis of reproducibility. Has the previous study used this data? The analysis seems to be very simple. The authors need to explain the results and show their meaning or significance to the field.
3. Figure 1 ONEST plots of different cut-off values for 23 pathologists were poorly plotted. There were no x-axis and y-axis. In Figure 1a, all the lines were interwind and no patterns can be seen.
4. Figure 2 had the same issues.
5. The analysis was very simple. Why so many authors were included? It's better to clearly state what contributions they all made to the results.
Author Response
RESPONSE Number 2 to REVIEWER 1 previously acting as REVIEWER 3
I am really sorry to note that our point-by-point answer to all the issues you raised and our list of revised items did not reach you, since we have done this very thoroughly, with the approval of all authors which took over a month. This is a complete and frustrating communication breakdown. You asked for major revision and we revised the manuscript accordingly, then we submitted the revised manuscript with the changes highlighted along with our reply, which did probably not get to you as you sent back the same concerns. I must say that the surface at the URL is not the most user-friendly one I have encountered, it took me quite some time to be able to get to your review (the same as the one we acted on before) again. It is also troublesome that the revised manuscript got another identification number, which made me contact the editor in charge, but she wrote back that this was normal…
We have highlighted all of our changes in red as you will see below (if ever this reaches you), but the editorial office has mandated to use the version they have modified and all highlights have been cleared and changes have been accepted, edited. I do not see much sense in deleting the new parts and reinserting them and they do not want us to use the previous version. It is very annoying that we might seem negligent due to the failure of the website to allow proper communication and transmit our changes and reply.
I am reproducing the comments to your initial (and repeated) review below, but this must be a system derived inconvenience to all of us and not negligence from our part. Again, the editorial system requires us to use the track changes of Word to highlight the changes, but previously highlighted changes have been incorporated into the manuscript version we are mandated to use. I think we are really innocent… In order to avoid similar lack of communication, I will try to copy and paste this at the initial pages of the manuscript, too.
The original revision notes follow below:
RESPONSE
We are grateful to Reviewer 3 for the time taken to review our manuscript and the comments provided. We have modified the manuscript to address these comments and we hope that the Reviewer will find the modifications satisfactory. Below is an explanation clarifying those changes.
- What is the difference between this study and the previous works, especially the ONEST analyses, using open-source software developed by the first author? There must be enough differences.
First, the open-source software was not available for the ONEST analyses of oestrogen and progesterone receptors, Ki67 or tumour histological grade. A prototype calculator was used that the developer alone could manipulate and not any of the remaining authors. The software was developed during this work to allow anyone to explore these aspects of reproducibility.
Second, while the first two studies reported on the ONEST analyses of reproducibility of (a) a simple parameter (quantification of nuclear staining for 3 different biomarkers) and (b) a more complex parameter (the combined histological grade being itself dependent on three other parameters) using the same series of cases, the present study evaluates sTILs, which has been shown to be a biomarker prone to poor reproducibility due to the complexity of its evaluation (heterogeneity, average required but without areas around normal structures, DCIS, previous biopsy site, necrosis… etc).
Third, the analysis presented here uses a different series with a dual analysis i.e. observer selected areas vs centrally selected areas evaluated in the same way and assessed by different observers. Only a third subset of cases overlaps with the tumours reported in the two previous ONEST reports, as the data were available although had not been reported previously and this allowed better comparisons of results that can be gained by ONEST. These parameters allowed us to study the method itself in a better way and to also report on its limitations. Indeed, limitations of ONEST- like influence by the number of observers, deviation from the majority (possibly relating to expertise) and heterogeneity were not previously explored or reported.
In summary, the difference between this report and the previous ONEST analyses is similar to the difference between several reproducibility studies evaluating different parameters i.e., the method is the same (like kappa or intraclass correlation coefficients that are used in most analyses), but the parameter assessed, the cases used, the observers, the methods of evaluation of the biomarker reported are different.
- The author used anonymized results from the EWGBSP analysis of reproducibility. Has the previous study used this data? The analysis seems to be very simple. The authors need to explain the results and show their meaning or significance to the field.
Reviewer 2 also asked more details about this, which suggests to us that the original description might not have been as straight forward as we felt it was. As stated in the first paragraph of the methods section, the results of the previous analysis (Reference 9) had been used. So, the answer to the first question is yes, and this has been stated in the manuscript. The same data were used but the analysis was different (ONEST vs intraclass correlation coefficient). The coefficient gave one approach to the reproducibility, and highlighted that heterogeneity was one of the main factors in lower agreement. ONEST itself validated the number of observers being sufficient for conclusions to be drawn from the date, but besides, the main contribution to the field, is probably that ONEST itself may yield different results depending on a number of factors and therefore it should also be cautiously interpreted. Indeed, the reviewer’s perception is correct, the analysis is not difficult, provided the permutations can be generated and their results made ready for analysis. It is this simplicity, and the added value of visualization and of the few parameters, that we consider to be relevant (bandwidth, ONEST value, OPA(n)) and which may make this analysis a good complementary tool for these types of reproducibility studies.
Some previous analyses of reproducibility for TILs (references 23-25) have reported high rates of concordant classifications, but their number of observers was well below the values we gained with ONEST. In the discussion we explain that these might be overestimations of reproducibility. We have added to the end of the discussion that simple ONEST analyses may estimate the reliability of published reports.
- Figure 1 ONEST plots of different cut-off values for 23 pathologists were poorly plotted. There were no x-axis and y-axis. In Figure 1a, all the lines were interwind and no patterns can be seen.
We regret that the reviewer has not liked the plots, which we felt were informative. Although the axes were not highlighted, they were labelled, and we probably incorrectly considered that this was sufficient. On the basis of the reviewer’s comment, we have corrected the lack of drawn axes by adding both x and y axes to all plots. As the reviewer felt the values are not easy to read from the originals, we have also added a vertical line projection to improve the reading of ONEST values. These are also represented in the first column of Table 1. An explanatory text for these vertical segments has also been added to the figure legend: “(this is highlighted by vertical segments between the x axis value and the minimum curve)”. We agree that when 100 significantly overlapping curves are represented in a single diagram, following the points of an individual line becomes difficult or impossible, even with 100 shades of colours. However, it is not the intention to read each individual OPAC from the diagram because the bottom points of this combination give the ONEST read, and Figure 1a is just an illustration for what Figure 1b is derived from, the latter includes all information we deem important from Figure 1a.
- Figure 2 had the same issues.
Similar to the response to comment 3 above, we have made the same modifications regarding the axes.and the vertical segments helping the reading of the values. The Figure is also there to illustrate the overlap and the overlap is also visualized.
- The analysis was very simple. Why so many authors were included? Have they all actually contributed to the results?
The contribution of each author is listed in the author forms signed individually. This has been updated in the manuscript file with a few errors corrected. To be able to analyze the data (even if the analysis is simple), the data need to be produced. The 23 pathologists involved in the first analysis have produced the data of series C1 and C2, whereas the 9 pathologists (one overlapping with the 23 mentioned before) produced the data of C3. These authors had to spend a considerable amount of time screening and scoring each slide for each circulation and reporting the values. Data management and curation also required considerable amount of time from other authors. In addition, the present analyses themselves were done by two authors, the first and last, whereas statistical advice, computation and control was done by an additional author. All have read the draft manuscript and commented on it. Therefore, all have contributed to the present work. When comparing the author list with that of the previous EWGBSP analysis (reference 9), only one of those authors (Davood Roshen) has not been included as a contributor to the present work, as he was not involved in data generation, curation and management, but in an analysis, which was not required for the present work. Such a high number of authors are not alien to reproducibility studies, e.g. the report on the TIL ring studies by the International Immuno-Oncology Biomarker Working Group or the one related to TILs in the IVITA study had both 41 authors (PMID: 27363491 and 34218258, respectively), two reports on PD-L1 reproducibility had 21 and 24, respectively (PMID: 34232604 and 31196152).
Reviewer 2 Report (New Reviewer)
Introduction provides sufficient information about the state of the field including enough references, and all of them are relevant to the research.
Research design is appropriate and methods are adequately described, including some key references.
Results are clearly presented and support the conclusions of the study, only 7-8 observers are enough to make a good estimation of the variability between them. Graphics are easy to understand.
The study is nolvety and has interest to scientific readers because it is a new way to analyse the inter-observer variability, helping to estimate the minimum number of observers required for reliable estimations in the evaluation of histopathology specimens of tumour infiltrating lymphocytes.
This manuscript has scientific soundness and a high quality presentation so in my opinion it merits to be published in Cancers.
Author Response
Thank you very much for the appreciation of our revised manuscript, which we think you have seen for the first time.
Round 2
Reviewer 1 Report (Previous Reviewer 3)
The authors have answered my questions.
It can be accepted in current form.
This manuscript is a resubmission of an earlier submission. The following is a list of the peer review reports and author responses from that submission.
Round 1
Reviewer 1 Report
Minor English spelling review needed.
Appreciated the honesty of the research in regard with the statistical significance.
Reviewer 2 Report
This paper proposes to quantitatively evaluate the level of TILs in the tumor based on the evaluation of TILs level by ONEST, to improve the reproducibility of STILS evaluation in breast cancer. The work of this paper is practical and logical. However, there are some problems to be further improved as well: 1 Brief summary and abstract: a good description, focusing on the background, objectives, methods, main findings and conclusions, please add a sentence explaining the necessity of the research. 2 The introduction is concise and understandable, using appropriate text, and presents the purpose of the study. 3 The logic of the method is clear, but the author should provide more specific data sources for this study. 4 The picture of the result is logical and the diagram is clear. 5 Discussion mentioned the main findings and recommendations based on the results, but I suggest explaining the innovative nature of this research. Does this article fill some knowledge gaps that have not been addressed by previous studies?
Reviewer 3 Report
The authors want to evaluate stromal TILs (sTILs) quantification by ONEST and to estimate the number of observers needed for a reliable evaluation of its reproducibility. There are several problems:
1. What is the difference between this study and the previous works, especially the ONEST analyses, using open-source software developed by the first author? There must be enough differences.
2. The author used anonymized results from the EWGBSP analysis of reproducibility. Has the previous study used this data? The analysis seems to be very simple. The authors need to explain the results and show their meaning or significance to the field.
3. Figure 1 ONEST plots of different cut-off values for 23 pathologists were poorly plotted. There were no x-axis and y-axis. In Figure 1a, all the lines were interwind and no patterns can be seen.
4. Figure 2 had the same issues.
5. The analysis was very simple. Why so many authors were included? Have they all actually contributed to the results?